# Numerical Analysis on the Structure Design of Precast Cement Concrete Pavement Slabs

Shuangquan Jiang [1,2], Yuan Wang [1], Xuhao Wang [1], Zexin Liu [1], Qianqian Liu [1], Cheng Li [1,*] and Peng Li [1]

1   School of Highway, Chang'an University, Xi'an 710061, China; jiangsq5311@126.com (S.J.); wangyuan@chd.edu.cn (Y.W.); wangxh@chd.edu.cn (X.W.); liuzexin1021@163.com (Z.L.); qqliu@chd.edu.cn (Q.L.); lipeng2013@chd.edu.cn (P.L.)
2   Sichuan Road and Bridge (Group) Co., Ltd., Chengdu 610000, China
*   Correspondence: cli@chd.edu.cn; Tel.: +86-18292898823

**Abstract:** The performance of cast-in-place cement concrete pavement can be greatly influenced by the surrounding environment and the quality of construction, and it requires longer curing time before opening to the traffic. The precast cement concrete pavement has the potential to address the disadvantages of using the cast-in-place cement concrete as the pavement surface material, and it also provides an alternative solution for the rapid repair of pavement damage. The current study aimed to assess the influences of geometry of full-scaled precast concrete slabs with various load transfer joint types. A numerical analysis was conducted using the finite element method to analyze the mechanical responses of the single and double slabs in accordance with the elastic thin-plate theory and the Specifications for Design of Highway Cement Concrete Pavement. Under the premise of determining the most unfavorable load position for different mechanical parameters, the mechanical response regularity under the critical loads was determined. The precast concrete pavement slab with dimensions 4 m long, 3 m wide and 0.28 m thick exhibited better mechanical responses when considering the maximum deflection and flexural stress as the evaluation index. The results indicated that the variation of joint forms has marginal influences on the maximum deflection and displacement transfer coefficient. When considering the maximum flexural stress and load transfer coefficient of the precast slabs, the circular tongue-and-groove joint form was recommended for application.

**Keywords:** precast concrete pavement; structure design; numerical analysis; finite element method



## 1. Introduction

Cement concrete pavement has the advantages of high strength, good durability, small economic cost and lower maintenance cost [1–4]. However, some shortcomings of the traditional cement concrete pavement have emerged over the decades of engineering practice and application, such as the long-time curing before opening to the traffic, and the complexity of workmanship, construction and maintenance. A prefabricated cement concrete pavement structure is put forward to effectively solve the problems of traditional cast-in-place cement concrete pavement. A precast concrete pavement (PCP) system is a set of specific panel details, materials and installation methods used in concert to rapidly create a fully functioning concrete pavement and has the potential to be recycled and reused. Well-designed PCP systems (i.e., structure and materials) enable rapid replacement with minimal impact on traffic flow and offer the potential for extended service life. These attributes are especially valuable in meeting pavement maintenance and repair needs in heavily trafficked areas and emergency rescue applications [5–9].

Over the years, PCP has proven itself useful in multiple applications. The varied applications of PCP that have been explored are: (1) intermittent repair of concrete pavement; (2) continuous construction or the new construction of concrete pavement; (3) airfield repair and construction; (4) temporary concrete pavement [10]. Precast slabs are the most widely used in repairing damaged roads, and precast repairs are sometimes loosely classified

as intermittent, continuous intermittent or intermittent and continuous replacement in adjacent lanes, but all three are similar in details and are frequently used in the same project. Approximately two-thirds of precast pavement projects constructed to date in the U.S. and Canada can be categorized as intermittent repairs, where "isolated" areas of the roadway have been replaced. Continuous full-lane replacement is appropriate when long stretches of any given lane have failed beyond the reasonable use of intermittent repair techniques. In addition to intermittent and continuous maintenance in a single lane, maintenance may also be performed in two or more adjacent lanes such that new precast panels may abut new precast panels or existing pavement in adjacent lanes.

Compared with traditional concrete pavement, assembled concrete pavement also has the advantages of saving raw materials and significantly reducing construction waste. At the same time, in the production of assembled concrete pavement, slabs can also use some new environmentally friendly materials to achieve the purpose of reducing carbon emissions and energy consumption [11–13].

However, limited literature has reported on the PCP system construction of all-paved roads and airport roads. In order to prompt the application of a continuous PCP system, it is necessary to optimize the critical parameters of PCP, such as full-lane panel dimensions and load transfer mechanisms (LTM), from the structure design stage before large scale applications. Liu [14] investigated the change of stress and deflection of different sizes of slab under the static load. The results showed that the thickness of the slab had less influence on the stress and deflection caused by the load, and that the thickness of the prefabricated slabs could meet the load-bearing capacity requirements with the same thickness of the old road slab. Luo et al. [15] investigated the effects of slab plane size on the stress and deformation in the slab at different substrate types, indicating that the deformation value increased to different degrees with the decrease in slab size regardless of the substrate. When the equivalent resilient modulus of the top surface of the substrate of small slabs (0.5 m × 0.6 m) was six times different, the stress difference was only 0.64%, which demonstrated that the adaptability of small slabs was better. Precast concrete panel repair technology research to determine the precast panel plane size and thickness was also conducted [16]. Through simulation analysis, it was proposed that the effects of slab dimensions on deflection were not as significant as those of flexural stress at the bottom of the slab.

In terms of joint forms between adjacent slabs, current state-of-art applications can be categorized into three modes: (1) directly filling the joint with gravel or flexible materials to provide the load transfer function of adjacent slabs; (2) the use of keyway joints, which transfer stress and strain through the tongue and groove; (3) connecting the adjacent slabs in the form of force transmission rod [17–19]. Tian et al. [20] optimized the trapezoidal groove joint of square concrete slabs in the size of 1 × 1 × 0.2 m, and drew the conclusion that the joint has the best load transfer ability when the tenon length is 4 cm and slope is 1:4. A similar conclusion was drawn by Zhao et al. [21]. In recent years, Yan [22] studied the stress–strain response of the slab with circular tongue-and-groove joints and found that the load transfer effect was optimal when the span vector ratio was two. Similarly, Zhang [23] studied the high-strength pervious precast concrete pavement and proposed a load transfer scheme with tongue-and-groove as the connection mode. The keyway joints were field demonstrated in the I-57 PCP system application in Missouri. On-site monitoring indicated that when the full depth of the keyway was not closely combined, the load transfer coefficient between slabs decreased sharply. This form was also used in the hexagon prefabricated concrete pavement developed in France [24]. Sadeghi [25] studied the influences of different load transfer modes on prefabricated pavement using the finite element software, ABAQUS, which provided a good preliminary investigation for this study.

Even though the concept of the PCP system has been developed for years, the main applications still focus on block pavement, small precast slabs and the repairing technology of cast-in-place concrete slabs. In order to optimize the design of a fully paved PCP system,

the study of prefabricated cement concrete pavement structure is divided into three phases: First, the numerical simulation stage. Second, the scale down test phase (according to the similarity theory to reduce the size of the slab until it can be done in the room). Third, the field test phase. The scaling test is currently in progress and further data processing is required. Subsequently, the scaling test data will be compared with the numerical simulation and appropriate adjustments will be made to continue the field test. The study in this paper focused on the numerical simulation stage which is the basis of subsequent studies. This study selected a wide range of full-scaled slab dimensions according to the current design code for cement concrete pavement [26]. The maximum flexural stress and maximum vertical displacement of the slab were taken as evaluation indices to recommend the appropriate slab structure. In order to ultimately reduce the life-cycle cost of the PCP system by recycling the full-scaled slabs for sustainability purposes, it was necessary to assess the load transfer coefficient of the innovative joint forms of slabs that could be easily assembled and disassembled. A variety of joint forms were proposed and the joint form with the best load transfer and with convenient assembly and disassembly abilities was recommended. The current study aimed to provide a theoretical reference for full-scaled PCP system engineering practice.

## 2. Experimental Design and Numerical Model

### 2.1. Theoretical Basis

In view of the research progress of precast cement concrete pavement in China, this research mainly optimized the design of the structure dimensions and joint forms of the pavement slabs. Based on the Specifications for Design of Highway Cement Concrete Pavement [26], the dimensions for the slab length, width and thickness were selected in a wide range, and five joint forms were proposed for the load transfer abilities evaluation. Firstly, the critical loading positions where the maximum flexural/tensile stress at the bottom and the maximum deflection of slab were determined. Secondly, selected slab dimensions—that can satisfy all the mechanical response indices when the critical loads were applied to the single slab at different positions—were determined. Thirdly, the critical loading positions for different joint forms, wherein the load transfer coefficient and displacement transfer coefficient, respectively, were determined. Finally, a combination of a reasonable slab dimension and joint form that met the specification requirements was recommended.

### 2.2. Design of Test Scheme and Establishment of Model

#### 2.2.1. Design of Size Optimization Scheme

As stated in the specification: a rectangular form should be adopted for the plane layout of ordinary cement concrete. The longitudinal and transverse joints should intersect vertically, and the transverse joints on both sides of the longitudinal joints should not misplace each other. In addition, the width, length and length–width aspect ratio should be selected within the range of 3.0–4.5 m, 4–5 m and 1–1.35 m, respectively, while the slab surface area should not be greater than 25 m$^2$ [26].

This study investigated the effects of different dimensions on the mechanical properties of a prefabricated pavement slab structure when selecting a number of sizes according to the size range specification. In order to assess the variation rule of various mechanical indices of slabs under different factors and different levels of action, this study simplified the size scheme with an orthogonal experimental design method. Using this method, a final number of sixteen orthogonal experimental schemes, with three factors and four levels of each factor, were set [27]. The test matrix is shown in Table 1.

**Table 1.** Design of the orthogonal test matrix of slab size.

| Size | 1 | 2 | 3 | 4 | 5 | 6 | 7 | 8 |
|---|---|---|---|---|---|---|---|---|
| Length | 4.0 | 4.0 | 4.0 | 4.0 | 4.5 | 4.5 | 4.5 | 4.5 |
| Width | 3.0 | 3.5 | 4.0 | 4.5 | 3.0 | 3.5 | 4.0 | 4.5 |
| Thickness | 0.22 | 0.24 | 0.26 | 0.28 | 0.24 | 0.22 | 0.28 | 0.26 |
| Size | 9 | 10 | 11 | 12 | 13 | 14 | 15 | 16 |
| Length | 5.0 | 5.0 | 5.0 | 5.0 | 6.0 | 6.0 | 6.0 | 6.0 |
| Width | 3.0 | 3.5 | 4.0 | 4.5 | 3.0 | 3.5 | 4.0 | 4.5 |
| Thickness | 0.26 | 0.28 | 0.22 | 0.24 | 0.28 | 0.26 | 0.24 | 0.22 |

### 2.2.2. Design of Joint Optimization Scheme

The convenience of factory prefabrication increases the varieties of the choice of joint forms for a full-scaled PCP system. Therefore, different forms of tongue-and-groove joints can be designed and assembled in the factory by using prefabricated joint molds in advance.

In general, the literature has categorized the joint forms into two styles—one with dowels and the other without dowels [28]. The joint form without dowels mainly includes a trapezoidal tongue-and-groove joint and a circular tongue-and-groove joint. In order to make a more comprehensive comparison and optimization of the joint form, this study added two joint forms—the inclined joint and the stepped joint, as exhibited in Table 2. In addition, the effects of different joint forms on the load transfer capacity between slabs were compared with a dowel bar insertion.

**Table 2.** Four types of joints without force transfer.

| Joint Form | Trapezoidal Tongue-and-Groove Joint | Circular Tongue-and-Groove Joint | Inclined Joint | Stepped Joint |
|---|---|---|---|---|
| Legend |  |  |  |  |
| Character | The gradient is 1:4 and the length is 4 cm | The ratio of span and rise is 2.0 | The slope is 45° | The form is a stage form |

Previous studies proposed that the load carrying capacity was optimal when the trapezoidal tongue-and-groove joint had a gradient of 1:4 and a length of 4 cm, which were applied in current study [20,29]. The cross-vector ratio of circular tongue-and-groove joints was 2.0 because the slab had the largest load transfer coefficient. The joint had the best load transfer capacity in this case [22]. The selection of inclined joints and stepped joints mainly referred to the form of the slab joints in bridge engineering, in which the slope of the inclined joints was 45°. The schematic and detailed setup for four joint forms without dowels are shown in Table 2. For the joint form with the dowel bar insertion, the spacing between the force transmission rods was chosen to be 0.3 m according to practical experience. Considering that the slab width was 0.3–0.45 m, the joint form with dowels was unified with nine dowels.

### 2.3. Material Design Parameters

A finite element numerical simulation method was adopted, and the simulation software (ABAQUS 2019), was introduced [25].

The following principles were proposed during the simulation process: (1) the cement concrete pavement slab is a small deflection elastic sheet with uniform material, as specified in the design specification; (2) the subgrade under the surface slab is set with the equivalent Winkler foundation model, and the "Elastic Foundation" contact form in contact analysis is used to simulate the Winkler foundation model below the subgrade [30]; (3) the plane size of the base layer and the foundation model are both larger than that of the pavement slab, and it is assumed that they satisfy the infinitely large foundation model specified in

the specification; (4) the interlayer contact mode is completely continuous, and the constraint type of the structural layer is simulated according to the actual constraint form [31]. The material design parameters and model of each structural layer are shown in Figure 1 and Table 3 [32].

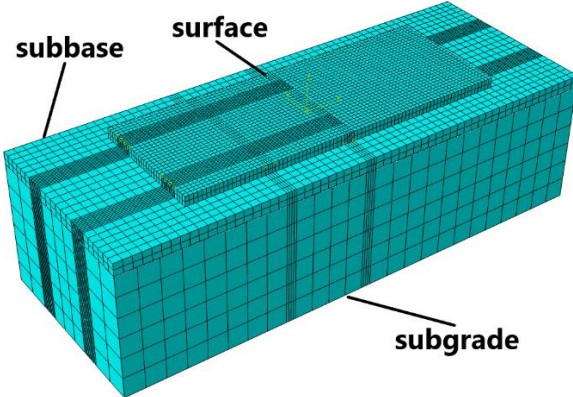

**Figure 1.** The structure layer model of the precast concrete pavement.

**Table 3.** Material design parameters of each structural layer in numerical simulation.

| Type | Length to Width (m) | Thickness (m) | Elasticity Modulus (MPa) | Poisson Ratio |
|---|---|---|---|---|
| Surface | Variable value | Variable value | 2500 | 0.15 |
| Subbase | 12 × 4.5 | 0.3 | 500 | 0.25 |
| Subgrade | 12 × 4.5 | 3 | 30 | 0.4 |
| | Length (m) | Radius (mm) | Elasticity modulus (MPa) | Poisson ratio |
| Dowels | 0.45 | 16 | 20,000 | 0.3 |

*2.4. Selection of Load Action Position and Critical Load Position*

The type of load action adopted in this simulation is the standard axle load of uniaxial double wheel group. According to the principle of load stress equivalence, the uniformly distributed load of the double circle can be converted into the rectangular load to simplify the meshing [33]. The axle weight is 100 KN, and the pressure of pneumatic tire is 0.7 MPa. The length–width ratio of the tire is 1.44. After conversion, the tire size is 0.23 m × 0.16 m, and the load action diagram is shown in Figure 2 [30].

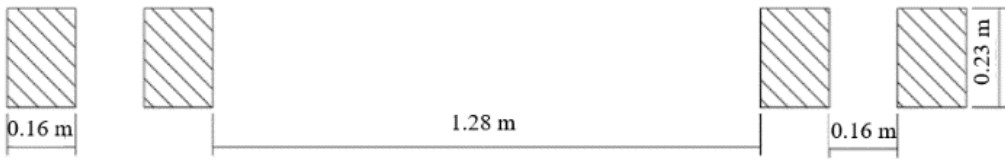

**Figure 2.** The graphic illustration of the standard axle load.

Based on the research of traditional precast pavement, it was found that the critical load position selected by the previous pavement research for different indicators was concentrated in a certain position. The most unfavorable position of the load is traditionally on the middle of the longitudinal joint; however, it is inconsiderate to select the same position as the critical load position for different indicators. This study selected different critical load positions for different indices. Three kinds of critical loads were mainly considered: a single axle double wheel set standard axle load was applied to center, the edge of the slab and the corner of the slab, respectively [30], which are shown in Figure 3.

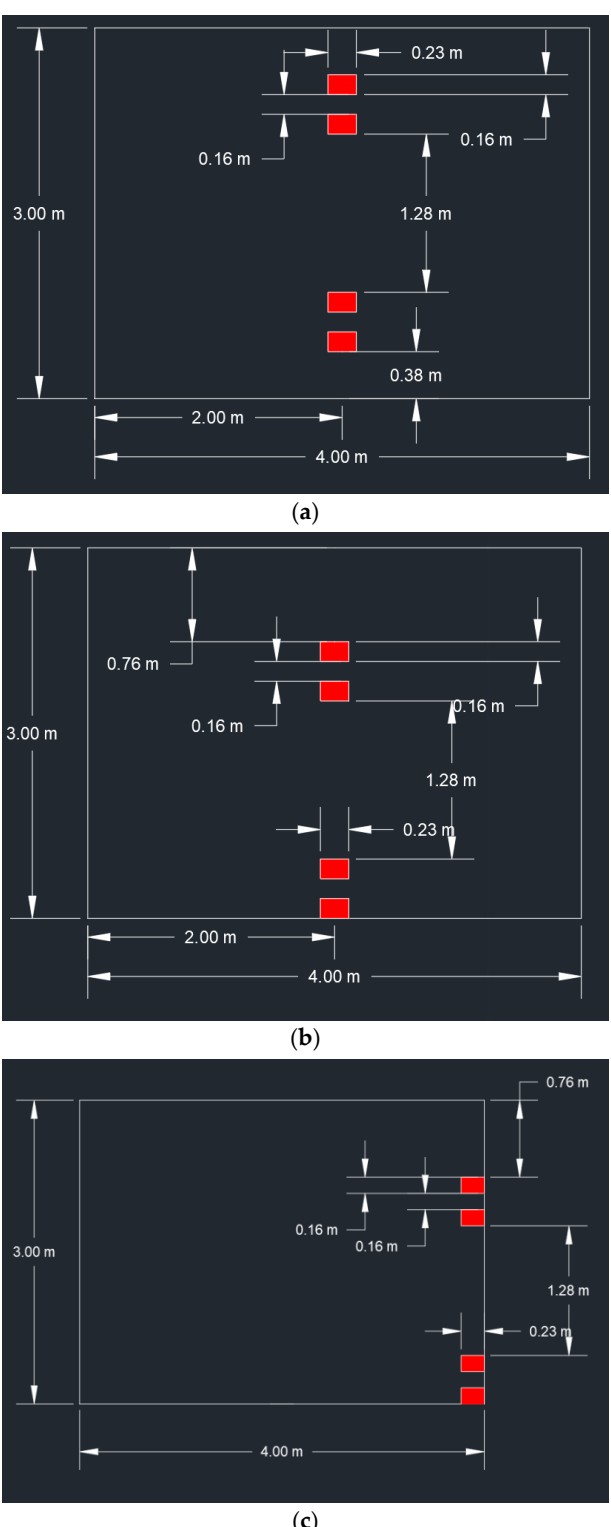

**Figure 3.** The load position as applied to different critical load position. (**a**) Middle; (**b**) edge; (**c**) corner (unit: m).

According to the design results of the orthogonal experiment, 16 different size combinations were carried out under the static load of three critical load positions. The numerical simulation was carried out to determine the most unfavorable load positions of the different indices. Each response index under the action of the most unfavorable load position was taken as the evaluation and optimization index to carry out the optimization design.

## 3. Optimization of Slab Size

Selection the evaluation indices: (1) Maximum flexural stress in the slab—the slab flexural damage standards in the surface layer thickness design also apply in this study, which is no fatigue fracture and limit fracture. The current design standards only consider the influence of thickness on the maximum flexural stress and does not consider the influence of plane size. Therefore, the influence law of different size variables on the flexural stress is considered in this study. (2) Maximum vertical deflection value—the critical control point of vertical shear failure is the maximum deflection change of the surface layer. Take the size dimensions of 4 m × 3 m × 0.22 m and the load applied to the middle of the single slab as an example. The mechanical response cloud diagram of the three evaluation indices is shown in Figure 4.

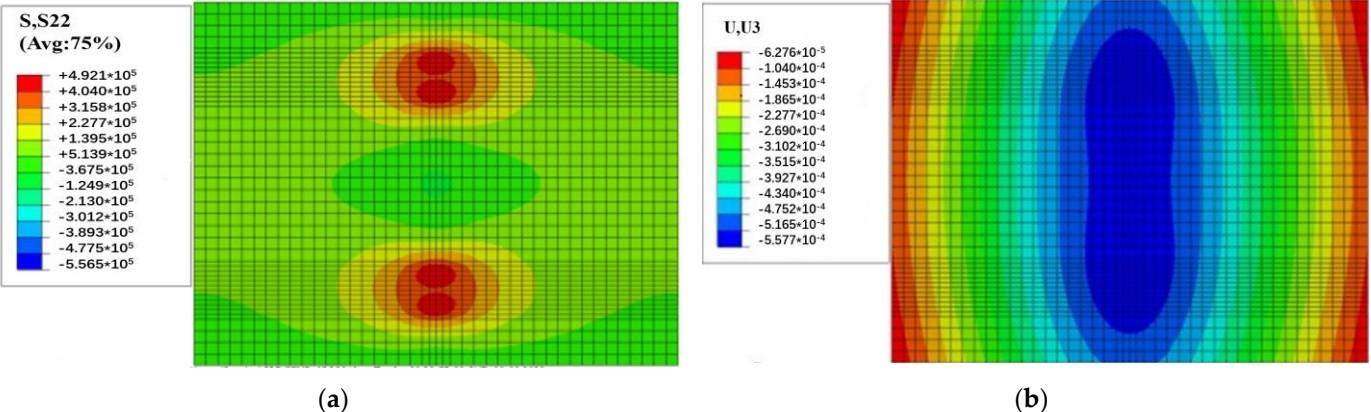

(**a**)                    (**b**)

**Figure 4.** Mechanical response nephogram of the load position for the middle. (**a**) Flexural stress in the slab; (**b**) vertical deflection value.

### 3.1. The Determination of the Critical Load Position of Different Indicators

3.1.1. Determination of Critical Load Position of Maximum Deflection Value

The influence of size on the maximum deflection value of the slab bottom is shown in Figure 5, which shows the standard axle load effect in center, edge and corner, respectively. The maximum deflection value emerges on the corner and the minimum deflection value emerges on the center under the same size structure by comparing the maximum deflection under different sizes of a single slab. Therefore, the most unfavorable load position of the slab's maximum deflection value is the load acting on the slab corner.

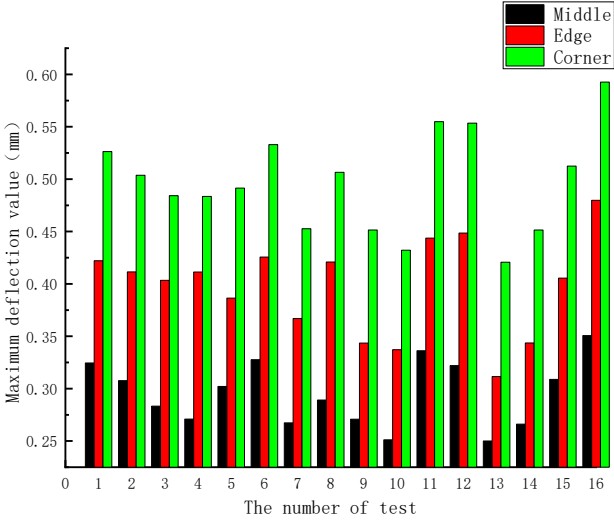

**Figure 5.** The influence of load position on the maximum deflection value.

### 3.1.2. Determination of Critical Load Potential of Maximum Flexural Stress

The influence of size on the maximum flexural stress of a slab is shown in Figure 6. Figure 6 shows that, by comparing the maximum stress under different sizes—during test numbers 1, 2, 5, 6, 11, 12, 15 and 16, as well as when the thickness was 0.22 m or 0.24 m—the maximum flexural stress reached the maximum under standard axle load on the slab edge. In test numbers 3, 4, 7, 8, 9, 10, 13 and 14, as well as when the slab thickness was 0.26 or 0.28 m, the maximum flexural stress reached the maximum under standard axle load on the slab corner.

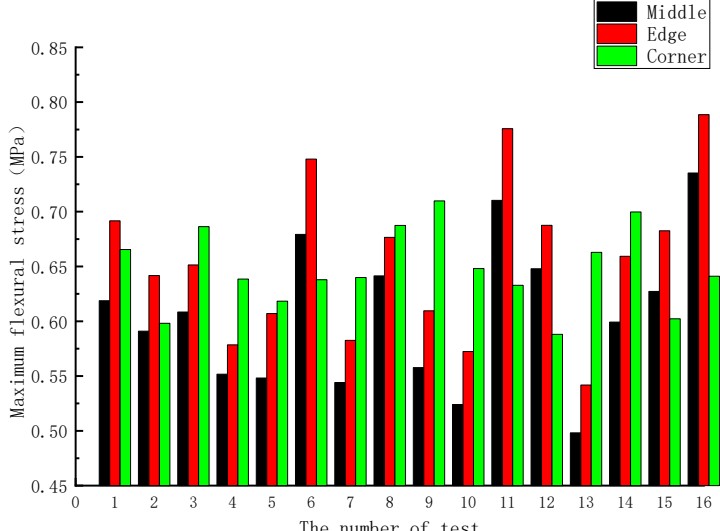

**Figure 6.** The influence of load position on the maximum flexural value.

In conclusion, when the thickness of the slab is 0.22 m or 0.24 m, the critical load position of the maximum flexural stress is selected as the edge of the slab. When the thickness of the slab is 0.26 m or 0.28 m, the critical load position of the maximum flexural stress of the slab bottom is selected as the slab corner.

### 3.2. Optimization of Slab Size under Different Critical Loads

3.2.1. Mechanical Response Analysis of Maximum Deflection Value

As stated above, when the maximum deflection of slab is used as the mechanical response index, the corner is considered as the critical load position to analyze the results. The average mechanical response indices of the slab maximum deflection value under different lengths, widths and thicknesses when the standard axle load of the uniaxial two-wheel assembly acts on the slab corner are shown in Table 4.

**Table 4.** Average mechanical response indices of the maximum deflection of slabs at different lengths, widths and thicknesses.

| Factors | Length | | | | Width | | | | Thickness | | | |
|---|---|---|---|---|---|---|---|---|---|---|---|---|
| The level of value (m) | 4.0 | 4.5 | 5.0 | 6.0 | 3.0 | 3.5 | 4.0 | 4.5 | 0.22 | 0.24 | 0.26 | 0.28 |
| Maximum deflection (mm) | 0.50 | 0.50 | 0.50 | 0.49 | 0.47 | 0.48 | 0.50 | 0.53 | 0.55 | 0.52 | 0.47 | 0.45 |

The influences of different factors such as length, width and thickness on the slab maximum deflection value are shown in Figure 7. It shows that when the maximum deflection of the slab is considered as the mechanical response index, the most unfavorable load position for maximum deflection value is the slab corner. By calculating the average value of different factors at different levels, the variation trend of the maximum deflection value with different sizes is determined.

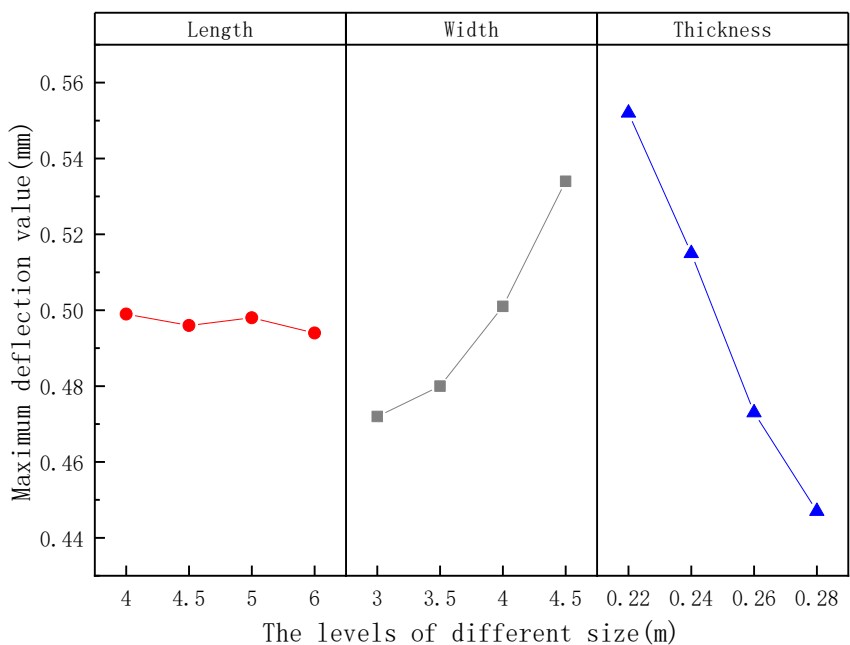

**Figure 7.** The influence of size on the maximum deflection value.

The results show that the maximum deflection value of the slab is almost unchanged with the increase of the slab length, so that the change of the length has little influence on the maximum deflection value. It increases simultaneously with the increase in the slab width, and the maximum deflection value reaches the maximum when the width is 4 m. It decreases with the increase in thickness, and the maximum deflection value of the slab is significantly smallest when the thickness of surface slab reaches 0.28 m.

To summarize, when the slab corner is taken as the most unfavorable load position of the maximum deflection index, four levels of surface slab length can be selected, and the width and thickness can be selected as 3 m and 0.28 m, respectively.

### 3.2.2. Mechanical Response Analysis with Maximum Flexural Stress

After analyzing the most unfavorable load position for maximum flexural stress, the positions in the edge and the corner are determined for the factor of length and width. In addition, the most unfavorable load positions are the corner, where the width is 0.22 m or 0.24 m, and the edge, where the width is 0.26 m or 0.28 m.

The mean mechanical response index of the slab maximum flexural stress under different lengths, widths and thicknesses when the standard axle load of the uniaxial two-wheel assembly acts on the unfavorable load position is shown in Table 5.

**Table 5.** Mean mechanical response indices of the slab maximum flexural stress values under different lengths, widths and thicknesses.

| Factors | | Length | | | | Width | | | | Thickness | | | |
|---|---|---|---|---|---|---|---|---|---|---|---|---|---|
| The level of value (m) | | 4.0 | 4.5 | 5.0 | 6.0 | 3.0 | 3.5 | 4.0 | 4.5 | 0.22 | 0.24 | 0.26 | 0.28 |
| Maximum flexural stress (MPa) | edge | 0.64 | 0.65 | 0.66 | 0.67 | 0.61 | 0.66 | 0.67 | 0.68 | 0.75 | 0.66 | - | - |
| | corner | 0.65 | 0.65 | 0.65 | 0.65 | 0.66 | 0.65 | 0.64 | 0.64 | - | - | 0.70 | 0.65 |
| | mean | 0.64 | 0.65 | 0.65 | 0.66 | 0.64 | 0.65 | 0.66 | 0.66 | 0.75 | 0.66 | 0.70 | 0.65 |

The influence of different size factors on the maximum flexural stress of the slab bottom is shown in Figure 8. It can be observed that the variation trend of the maximum flexural stress with the increase in size is determined by calculating the mean response index. The results show that the maximum flexural stress increases gradually with the increase in

length, but the variation range is not large, and the limiting value is only 0.016 mpa. The index also increases gradually with the increase in the width, and the change range is also small. The variation rule with slab thickness is obvious and the flexural value is smallest compared to the others.

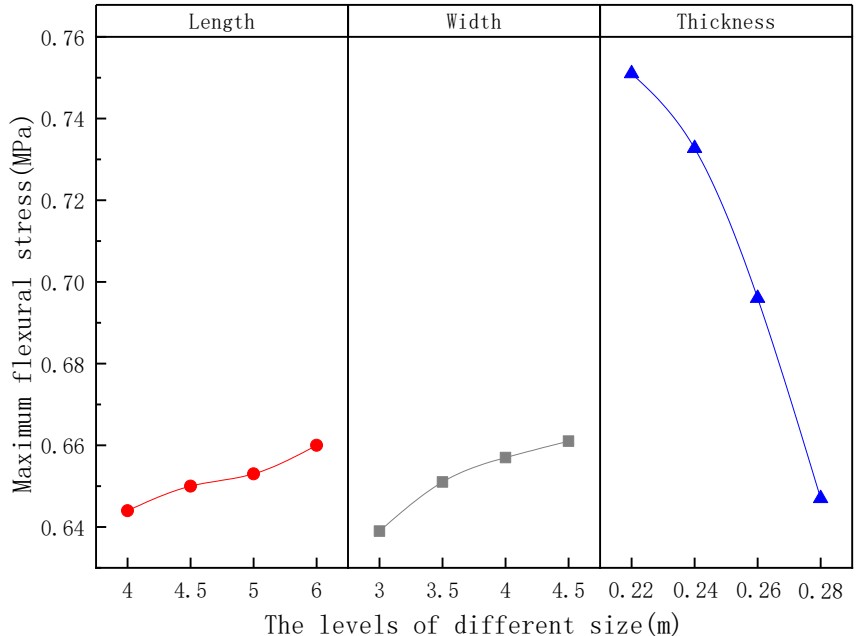

**Figure 8.** The influence of size on the maximum flexural value.

To summarize, when the maximum flexural and tensile stress values of the slab bottom are taken as the optimization indices, a smaller value can be appropriately selected in terms of length and width, and 0.28 m is recommended for thickness.

Combined with the above analysis results, it is suggested to select 4 m for the length, 3 m for the width and 0.28 m for the thickness of the slab.

### 3.3. Simulation Verification of the Recommendation

This numerical simulation test adopts the orthographic test design. Because the size combination of the final recommended size has not yet been simulated, this section mainly conducts the mechanical index verification under the most unfavorable load for these two size forms, as shown in Table 6.

**Table 6.** Mechanical response index verification of recommended size form in veneer form.

| Test Number | Length (m) | Width (m) | Thickness (m) | Edge | Corner | |
|---|---|---|---|---|---|---|
| | | | | Maximum Flexural Stress (MPa) | Maximum Flexural Stress (MPa) | Maximum Deflection (mm) |
| No.17 | 4.0 | 3.0 | 0.22 | 0.60 | - | 0.50 |
| No.18 | 4.0 | 3.0 | 0.28 | - | 0.65 | 0.44 |

By comparing experiments 17 and 18 with the former 16 groups, it was found that the size of No.18 by 2.2 combination in the most unfavorable load under the action of the mechanical response were close to the minimum.

It means that when the length is 4 m, width is 3 m and thickness is 0.28 m within the design size range specified in the specification, the occurrence of angular compression, bending, tensile failure and vertical shear failure of the slab can be effectively reduced.

## 4. Optimization of Joint Form between Two Slabs

Selection of the optimization index: The joint load transfer coefficient Kw is the ratio of the deflection value between the joint edge of the unloaded and loaded slab. It is determined that the joint load transfer capacity is optimal when it is greater than 80%. The optimization index of the joint load carrying capacity can be divided into two types: the displacement load transfer coefficient and the stress load transfer coefficient [34]. Equation (1) gives the displacement load transfer coefficient:

$$K_U = \frac{U_2}{U_1} \times 100\% \tag{1}$$

where $U_1$ is the maximum deflection of the edge of the slab under load, and $U_2$ is the maximum deflection of the edge of the slab under load. The stress load transfer coefficient is given as follows:

$$K_\sigma = \frac{\sigma_2}{\sigma_1} \times 100\% \tag{2}$$

where $\sigma_1$ is the maximum flexural stress value of the slab side under load, and $\sigma_2$ is the maximum flexural stress value of the edge of unloaded slab.

The two size forms are the 4 m slab length, 3 m width and 0.22 m or 0.28 m of thickness. The five joint forms are the inclined joint, stepped joint, trapezoidal tongue-and-groove, circular tongue-and-groove and joint with dowel steel. There are 10 combination forms that were designed from and five joint forms and two size forms to optimize the selection of different joint forms in the double slab form, as shown in Table 7.

**Table 7.** Joint form scheme design under double slab form.

| Test | 19 | 20 | 21 | 22 | 23 | 24 | 25 | 26 | 27 | 28 |
|------|----|----|----|----|----|----|----|----|----|----|
| Joint form | Inclined joint | | Stepped joint | | Trapezoidal tongue-and-groove | | Circular tongue-and-groove | | Joint with dowel steel | |
| Length | 4 | 4 | 4 | 4 | 4 | 4 | 4 | 4 | 4 | 4 |
| Width | 3 | 3 | 3 | 3 | 3 | 3 | 3 | 3 | 3 | 3 |
| Thickness | 0.22 | 0.28 | 0.22 | 0.28 | 0.22 | 0.28 | 0.22 | 0.28 | 0.22 | 0.28 |

The numerical simulation results are presented as follows: the size is $4 \times 3 \times 0.24$ m, and the position of loading is the middle of the slab. The mechanical response cloud diagrams of evaluation indices are shown in Figure 9.

### 4.1. The Relationship between the Load Transfer Coefficient and the Critical Load Position

4.1.1. Relationship between Stress Transfer Coefficient and Critical Load Position

The influence of different joint forms on the stress transfer coefficient is shown in Figure 10. By comparing the stress transfer coefficient with the different forms of joints, it shows that when the standard axial load is respectively applied to different positions of the slab, the coefficient value is significantly higher than the load acting on the corner. Selecting the corner is obviously not reasonable as an index for optimizing the form of the joints. When the load is applied to the middle or side of the slab, the stress load transfer coefficient has little difference under the same joint structure, and is generally less than the value of the corner.

In conclusion, when the stress transfer coefficient is used as the evaluation index, the edge or the middle of the slab can be selected as the critical load position.

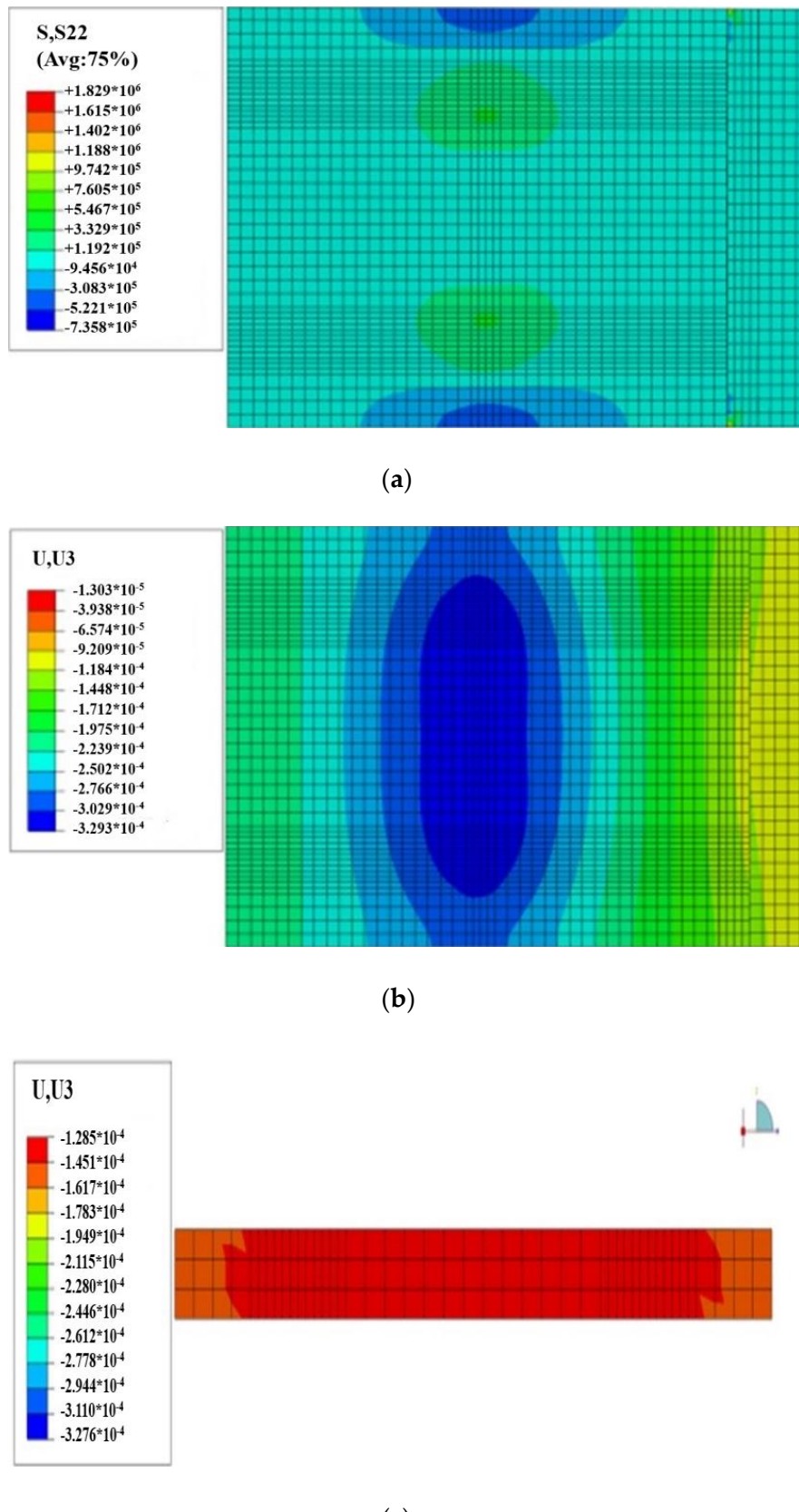

(**a**)

(**b**)

(**c**)

**Figure 9.** *Cont.*

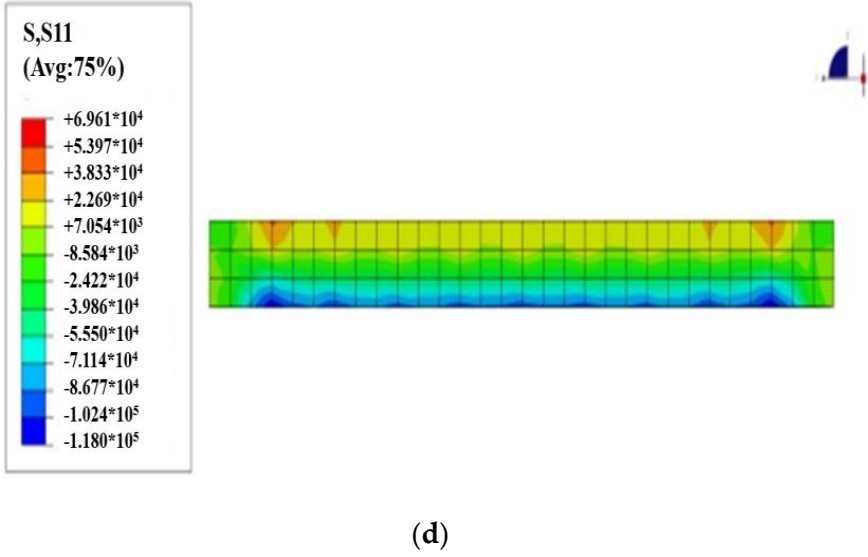

(**d**)

**Figure 9.** Mechanical response nephogram as the joint with dowel. (**a**) Maximum flexural stress of bottom; (**b**) the deflection value of bottom; (**c**) the displacement of unload slab; (**d**) the stress of slab under load.

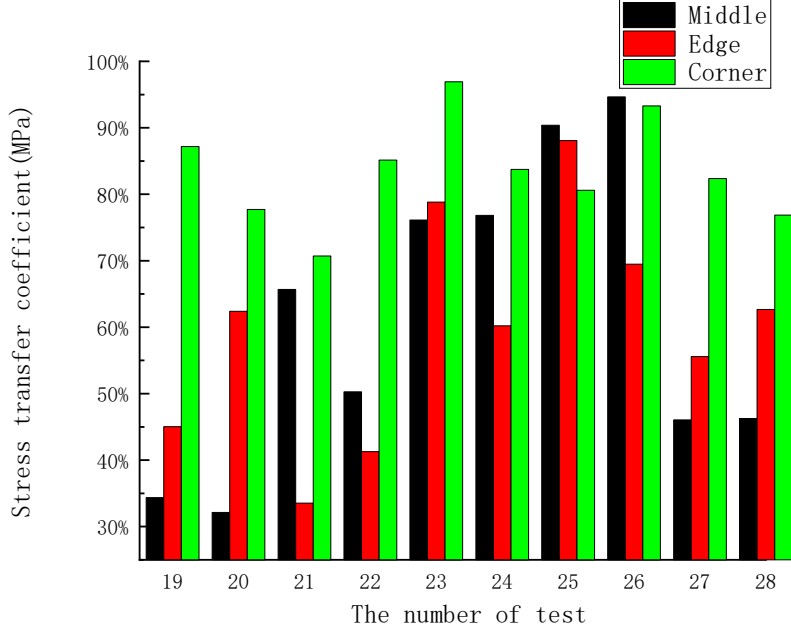

**Figure 10.** The influence of load position on the stress transfer coefficient.

4.1.2. Relationship between Displacement Transfer Coefficient and Critical Load Position

The influence of different joint forms on the displacement load transfer coefficient is shown in Figure 11. Through the comparison of the displacement load transfer coefficient of different load position under same joint forms, it can be observed that the displacement load transfer coefficient is at a maximum when the standard axle load acts on the middle of the slab, and it is at a minimum when the load acts on the corner.

To summarize, when displacement load transfer coefficient is selected as the evaluation index, the critical load position is the corner of the slab.

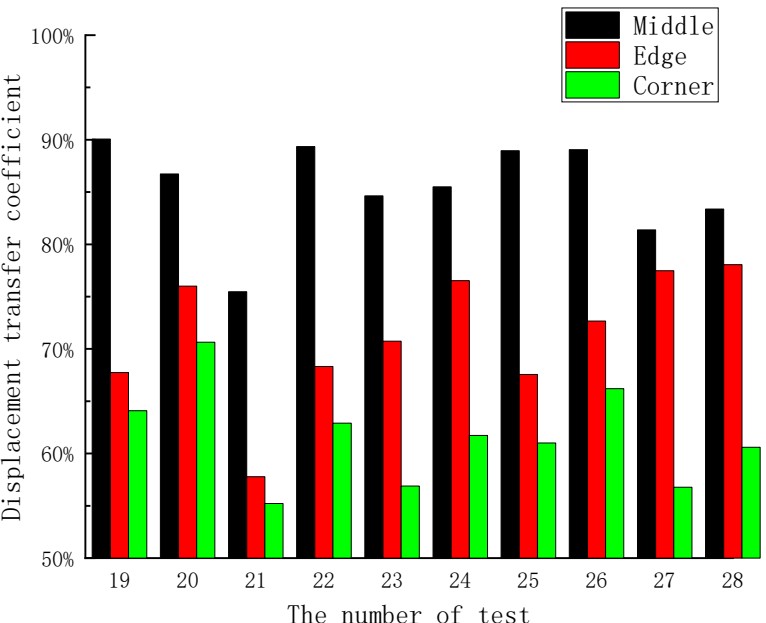

**Figure 11.** The influence of load position on the deflection transfer coefficient.

### 4.2. Optimization of Joint Form under Different Values and Different Critical Loads

From what has been discussed above, in this study, when the stress transfer coefficient value is used as the mechanical response index for analysis, the edge of slab can be selected as the critical load. When the displacement load transfer coefficient value is used as the mechanical response index for analysis, the corner of the slab is selected as the critical load position in this study.

The standard axial load of the uniaxial double-wheel set acts on the critical load position. The mechanical response mean indices under different forms, such as the maximum flexural stress, the maximum deflection value, the stress transfer coefficient and the displacement transfer coefficient, are shown in Table 8.

**Table 8.** Mean indices of different mechanical responses under different joint forms.

| Thickness | Joint Form | Maximum Flexural Stress (MPa) | Maximum Deflection (mm) | Stress Transfer Coefficient | Displacement Transfer Coefficient |
|---|---|---|---|---|---|
| The critical load position | | Edge (width is 0.22 m) Corner (width is 0.28 m) | Corner | Edge | Corner |
| | ① | 0.61 | 0.38 | 0.34 | 0.64 |
| | ② | 0.61 | 0.39 | 0.66 | 0.55 |
| Thickness of 0.22 m | ③ | 0.82 | 0.37 | 0.76 | 0.57 |
| | ④ | 0.75 | 0.35 | 0.90 | 0.61 |
| | ⑤ | 0.70 | 0.36 | 0.46 | 0.57 |
| The standard deviation | | 0.079 | 0.08 | 0.01 | 0.20 |
| | ① | 0.09 | 0.37 | 0.32 | 0.71 |
| | ② | 0.13 | 0.32 | 0.50 | 0.63 |
| Thickness of 0.28 m | ③ | 0.28 | 0.32 | 0.77 | 0.62 |
| | ④ | 0.27 | 0.30 | 0.95 | 0.66 |
| | ⑤ | 0.32 | 0.32 | 0.46 | 0.61 |
| The standard deviation | | 0.092 | 0.09 | 0.02 | 0.23 |

①—inclined joint, ②—stepped joint, ③—trapezoidal tongue-and-groove, ④—circular tongue-and-groove, and ⑤—joint with dowel steel.

The influence on different mean mechanical response indices of different joint forms is shown in Figure 12. When optimizing the joint forms, the maximum flexural stress of the slab bottom, the maximum deflection value, the stress and displacement transfer coefficient are mainly considered. Some observations are shown here:

(1) When the maximum deflection value and displacement transfer coefficient were taken as mechanical response indices, it was found that the fluctuation range of two values are less than 0.1, so that the variation of different joint forms had little impact on these indices. The results show that each of the joint forms can meet the requirements when the two indices are used as the control factors.

(2) When the maximum flexural stress of slab bottom was taken as the mechanical response index, the simulation results showed that the maximum flexural stress with the slab thickness of 0.22 m was higher than the value of 0.28 m. Under the condition of the same thickness, the maximum flexural stress of the inclined and stepped joints were obviously lower than the rest of joint forms. Therefore, the recommended thickness is 0.28 m, and the joints for the maximum flexural stress value were the inclined joints or stepped joints.

(3) When the stress transfer coefficient was taken as the mechanical response index, the stress transfer coefficient of the circular tongue-and-groove joint was much higher than that of other joints under the condition of the same thickness. The values reached 90.38% and 94.66%, respectively, for slab thicknesses of 0.22 m and 0.28 m. In this case, the circular tongue-and-groove joint was recommended for the stress transfer coefficient.

In conclusion, the inclined or stepped joints were chosen when the maximum flexural stress at the bottom of the slab was a control index, and the circular tongue-and-groove joint was selected when the stress load transfer coefficient was a control index. However, Figure 12 shows that stress transfer coefficient of the inclined and stepped joint only reached 60%, and they could not satisfy the requirement of 90% in the specification. Therefore, the circular tongue-and-groove joint is the best joint form and is recommended for the precast concrete pavement.

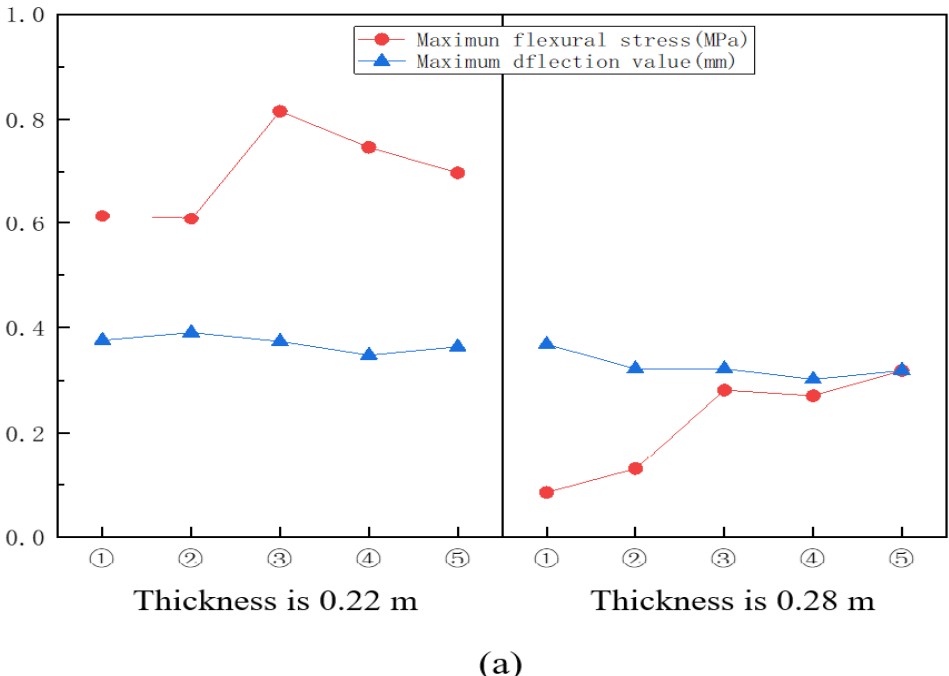

(a)

**Figure 12.** *Cont.*

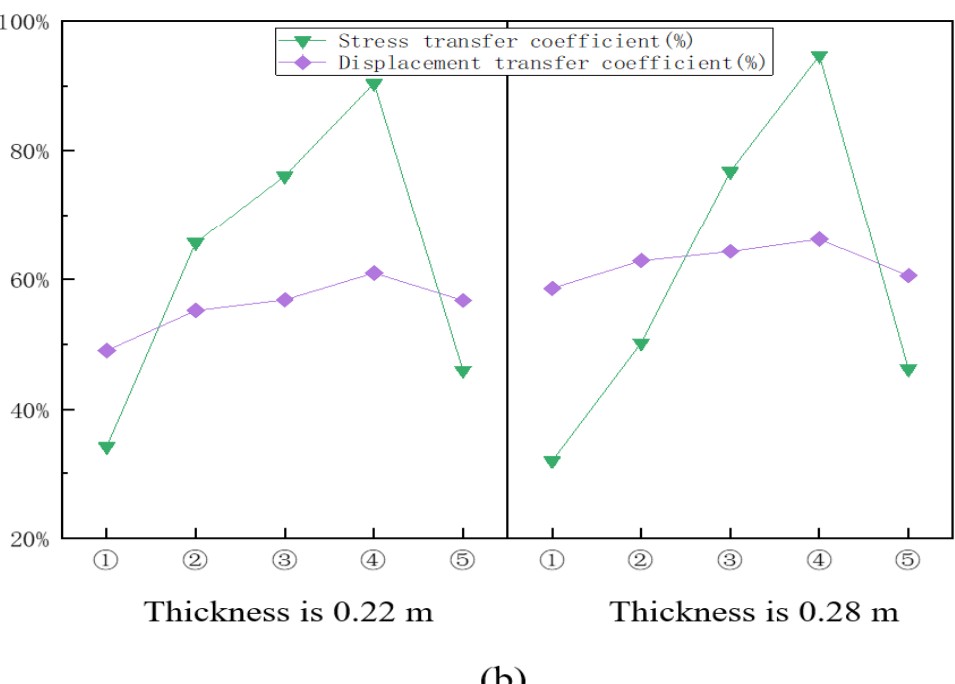

**Figure 12.** The influence of different joints on average mechanical response nephogram.(**a**) The maximum flexural stress and maximum flexural stress of deflection value for slab thicknesses of 0.22 m and 0.28 m; (**b**) the stress transfer coefficient and displacement transfer coefficient for slab thicknesses of 0.22 m and 0.28 m.(①—inclined joint, ②—stepped joint, ③—trapezoidal tongue-and-groove, ④—circular tongue-and-groove, and ⑤—joint with dowel steel).

## 5. Optimization Suggestions and Discussion

The load transfer capacity along the joints is a critical parameter for PCP design. Vahid [25] compared the load transfer capacity of the three joint forms considering the static load applied at the middle of the slab and using the load stress transfer coefficient as the single index to evaluate the load transfer capacity. In this study, the static load was applied to the edge, middle, and corner of the slab to simulate the potential vehicle load transfer effects at the joints. The simulation results indicated that the load transfer capacity of the joints was varied when the static load was applied to the different positions of the slab. In addition, both the displacement and load transfer coefficient were introduced as the joint load capacity indicators. It was found that when the slab thickness was smaller than 0.25 m, the corner of the slab should be selected as the critical loading position and the displacement transfer coefficient was taken as the evaluation indicator, whereas the edge and corner of the slab should be considered as the critical load positions. The load transfer coefficient was selected as the evaluation index for a slab thickness greater than 0.25 m.

The effect of slab thickness on the maximum vertical displacement and maximum flexural stress of the slab was greater than that of the slab length and width. The vertical displacement of the slab decreased as the slab thickness increased, in compliance with previous literature [17]. The optimization process of the PCP slab structure was derived from the finite element simulation and control indices were adopted in accordance with the Chinese specification of the JTG D40 "Design of Highway Cement Concrete Pavement" [26].

Comparisons of the load transfer effects of innovative joint forms have been presented in previous studies [22,23,29], and the keyway joint and the load transfer bar joint are the most recommended configurations to connect adjacent slabs. The current study provided comprehensive comparisons on joint forms and load transfer effects from different dimensions of the PCP slab system. Even though the circular tongue-and-groove joint had a lower load transfer coefficient compared to the dowel bar inserted joint form, the coefficient still satisfied the specification requirements of 90%. In order to ultimately reduce the life-cycle

cost of the PCP system by recycling the full-scaled slabs for sustainability purposes, it was highly recommended to apply the circular tongue-and-groove joint for fast assembly and disassembly purposes.

## 6. Conclusions

Based on the analysis and discussion presented in previous sections, the following conclusions which will provide a basis for the scale down test and the field test can be drawn:

(1) Within the range of slab sizes recommended by the current cementitious pavement design specifications, the optimal dimension of a full-scaled PCP system slab was 4 m long, 3 m wide and 0.28 m thick when the maximum flexural stress in the slab and the maximum vertical displacement of the slab were applied as evaluation indicators.

(2) The variation of the joint type had little influence on the displacement transfer coefficient. When the load transfer coefficient was used as the evaluation index of the load transfer effects, large variations were observed. The load transfer coefficient of the proposed circular tongue-and-groove joint form can satisfy the requirement of 90% in the specification. Therefore, the circular tongue-and-groove joint is the best joint form and is recommended for the precast concrete pavement.

**Author Contributions:** Conceptualization, X.W. and C.L.; data curation, X.W. and Z.L.; formal analysis, S.J. and Y.W.; investigation, S.J. and Y.W.; methodology, X.W. and C.L.; resources, X.W.; supervision, X.W., C.L. and P.L.; writing—original draft preparation, S.J. and Y.W.; writing—review and editing, X.W., Z.L., Q.L. and P.L. All authors have read and agreed to the published version of the manuscript.

**Funding:** This study was also partially supported by the National Natural Science Foundation Project (NSFC 51868066, 52178185), the Shaanxi Natural Science Basic Research Program, (Grant No. 2021JQ-249), the Scientific research project of China Railway first survey, the design and Research Institute (19–38), and the fellowship of China Postdoctoral Science Foundation, (Grant No. 2021MD703885).

**Institutional Review Board Statement:** Not applicable.

**Informed Consent Statement:** Not applicable.

**Data Availability Statement:** Not applicable.

**Conflicts of Interest:** The authors declare no conflict of interest.

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
