# Peer review of "Numerical Analysis on the Structure Design of Precast Cement Concrete Pavement Slabs"

_coatings, doi:10.3390/coatings12081051_

Round 1

Reviewer 1 Report

Dear authors,

In this paper, the authors present an analysis of  numerical and statistical modeling of structure design of precast cement concrete pavement slabs, which seeks an evaluation of the influences of its size and thickness.

As weak elements:

The introduction could be more consistent.

Figure 3 is a little bit concise - it can be completed with graphic elements that convey to the reader necessary information for a complete understanding of the paper. 

Figure 4 can be moved to the right so that it is not necessary to exit the format.

In figure 5 please check the ordinate with the explanations provided in the text. I think a mistake has been made.

Table 4 column 4 – there is an error in the header of the table.

Figure 9 - All images can be reduced to fit the format.

12 - In the legend above there is a mistake (the word deflection).

Optimization suggestions and conclusions - should remain only   ”Conclusions considering that the optimization has been done before”.

The bibliography must be extended to current papers and reach at least 30 references. At such a level it is not possible to present a research with such a small number of bibliographic references and which stops in 2019.

And in terms of notable elements

The authors present a very interesting and complete analysis with novelty elements using numerical modeling and statistical analysis.

In the future, I suggest to the authors to try  the ”Design of experiment method” in order to be able to optimize more efficiently the parameters from the numerical modeling.

Reviewer 2 Report

Paper is not suitable to be published in this journal. Providing and presenting basic parts of a research paper was missing. For example, in a good research paper, introduction is an important part where definitions, reviewing the literature and research gap should be well presented to justify why this research is needed. In addition, research is just in theory without validating the results.

Reviewer 3 Report

The manuscript entitled "Numerical Analysis on the Structure Design of Precast Cement Concrete Pavement Slabs" presents an interesting study conducted on the optimization of slab geometry through numerical simulation. However, the literature survey wasn’t conducted, the paper doesn’t have a discussion section and many other issues must be addressed. The paper needs major revisions before it is processed further, some comments follow:

·       The abstract is written qualitatively. The majority of the qualitative statements should be modified for quantified result comparisons. Currently, the abstract doesn't clearly show the results obtained in this research. Also, in the first sentence "can be used in special regions such as permafrost and seasonal-frost regions", then in the second "can be greatly influenced by the surrounding temperature" – the formulations are contradictory and ambiguous. Please revise.

·       Introduction. The first two paragraphs include many affirmations without a clear background in the literature. Please provide relevant citations/references or experimental results to support your affirmations.

·       Introduction. The introduction is way too short and the current level of research in the field wasn’t presented. Please evaluate suitable literature and clearly state their contributions, pros, and cons and how the current work would address them. Also, to avoid the "shortcomings" of cement-based materials a greener alternative was discovered and widely presented in the literature (consider these studies in the introduction: DOI: 10.1088/1757-899X/374/1/012068 DOI: 10.1515/epoly-2022-0015 and DOI 10.3390/ma15010202). The authors should provide a short paragraph where it can be said that the results of this study can/cannot also be applied to geopolymers.

·       Figure 2 – please introduce measuring units.

·       Figure 3 – please introduce the figure axis and increase the clarity of the presented images.

·       Determination of critical load position and critical load potential – Please provide some comments related to data distribution. Why is the difference so high between the tests? Also, is there a clear relation between the deflection values of the middle, edge, and corner? Please introduce relevant comments.

·       Discussion section. The discussion section is missing. In the discussions section, clear correspondence and comparison between the results of this study and those from the literature should be provided. Please improve.

Reviewer 4 Report

The manuscript “Numerical Analysis on the Structure Design of Precast Cement Concrete Pavement Slabs” studied the evaluation of the influences of the size and thickness of precast concrete slabs with various load transfer joint types. Τhe results are well presented. The authors have employed correctly the techniques. I have found the methodological approach correct. The presentation of the problem is clear, the results correctly presented and the conclusions well explained. English is not bad and generally is easy to follow, but there are some evident grammar mistakes. To conclude, I suggest this manuscript to be published in the journal “Coatings” after the below major revisions:

Ø  The introduction needs to be more emphasised on the research work with a detailed explanation of the whole process considering past, present, and future scope. The introduction is very brief, the authors should supplement it.

Ø  The purpose of the study needs better wording.

Ø  In some parts of the manuscript, authors discussed the results in light of the literature prior to presenting their results, which is distracting. Τhere is no separate section for discussing the results.

Ø  I think is in article the Authors presented only investigations, but what about technology?

Ø  The final question concerns for the economically aspects the process?

Round 2

Reviewer 2 Report

From my comments and the other reviewers, the authors' actions improved the paper. However, I believe that doing just simulation in this field is not a significant contribution. I leave it to the editor’s decision and respect the coming decision. However, I strongly advise that if it is going to be accepted this limitation must be highlighted in the main text (for example in research method) and at the end of conclusions. The limitations of study that highlighted in the authors’ response to my comment. The authors can highlight the 3 stages of study and emphasise that this research is in what stage.  

Reviewer 3 Report

The authors addressed most of my comments and the article was improved accordingly. The paper can be processed further.

Reviewer 4 Report

The authors carefully followed the comments and suggestions, made appropriate corrections and the manuscript in the present form was sufficiently improved with respect to the previous version. I recommend to accept this manuscript for publication.